# Complex plant-derived organic aerosol as ice-nucleating particles – more than a sum of their parts?

Isabelle Steinke[1,*], Naruki Hiranuma[2], Roger Funk[3], Kristina Höhler[1], Nadine Tüllmann[1], Nsikanabasi Silas Umo[1], Peter G. Weidler[4], Ottmar Möhler[1], Thomas Leisner[1]

1. Institute of Meteorology and Climate Research – Atmospheric Aerosol Research, Karlsruhe Institute of Technology, Karlsruhe, Germany
2. Department of Life, Earth and Environmental Sciences, West Texas A&M University, Canyon, USA
3. Working Group Landscape Pedology, Leibniz-Centre for Agricultural Landscape Research, Müncheberg, Germany
4. Institute of Functional Interfaces, Karlsruhe Institute of Technology, Karlsruhe, Germany

*now at: Atmospheric Sciences and Global Change Division, Pacific Northwest National Laboratory, Richland, USA

Correspondence to: Isabelle Steinke (isabelle.steinke@pnnl.gov), Ottmar Möhler (ottmar.moehler@kit.edu)

## Abstract

Quantifying the impact of complex organic particles on the formation of ice crystals in clouds remains challenging, mostly due to the vast number of different sources ranging from sea spray to agricultural areas. In particular, there are many open questions regarding the ice nucleation properties of organic particles released from terrestrial sources such as decaying plant material.

In this work, we present results from laboratory studies investigating the immersion freezing properties of individual organic compounds commonly found in plant tissue and complex organic aerosol particles from vegetated environments, without specifically investigating the contribution from biological particles which may contribute to the overall ice nucleation efficiency observed at high temperatures. To characterize the ice nucleation properties of plant-related aerosol samples for temperatures between 242 and 267 K, we used the Aerosol Interaction and Dynamics in the Atmosphere (AIDA) cloud chamber and the Ice Nucleation Spectrometer of the Karlsruhe Institute of Technology (INSEKT), which is a droplet freezing assay. Individual plant components (polysaccharides, lignin, soy and rice protein) were mostly less or similarly ice-active compared to microcrystalline cellulose, which has been suggested by recent studies as a proxy for quantifying the primary cloud ice formation caused by particles originating from vegetation. In contrast, samples from ambient sources with a complex organic matter composition (agricultural soils, leaf litter) were either similarly ice-active or up to two orders of magnitude more ice-active than cellulose. Of all individual organic plant components, only carnauba wax (i.e. lipids) showed a similarly high ice nucleation activity as the samples from vegetated environments over a temperature range between 245 and 252 K. Hence, based on our experimental results, we suggest to consider cellulose as being representative for the average ice nucleation activity of plant-derived particles, whereas lignin and plant proteins tend to provide a lower limit. In contrast, complex biogenic particles may exhibit ice nucleation activities which are up to two orders of magnitude higher than observed for cellulose, making ambient plant-derived particles a potentially important contributor to the population of ice-

nucleating particles in the troposphere, even though major uncertainties regarding their transport to cloud altitude remain.

## 1 Introduction

Ice formation in the atmosphere has a significant influence on the microphysical and radiative properties of clouds. At temperatures above 235 K, atmospheric aerosol particles may act as ice-nucleating particles (INPs) (Pruppacher and Klett, 2010; Vali et al., 2015). In mixed-phase clouds, immersion freezing is often the dominant ice nucleation mode (Hande and Hoose, 2017). Immersion freezing refers to a solid particle initiating ice formation inside a supercooled cloud droplet.

Over the past decades, many different particle types initiating freezing in mixed-phase clouds have extensively been studied (Hoose and Möhler, 2012; Murray et al., 2012; Kanji et al., 2017). Mineral dust particles emitted from desert areas have been identified as ubiquitous INPs which initiate ice nucleation in clouds over a wide range of temperature and humidity conditions (Boose et al., 2016; Ullrich et al., 2017). Cloud-level concentrations of potentially very ice-active primary biological aerosol particles (Hoose and Möhler, 2012) are much lower than background concentrations of mineral dust, with differences of up to 8 orders of magnitude in some cases (Hummel et al., 2018). Nevertheless, several laboratory studies, remote sensing measurements, and studies characterizing ice crystal residuals have found evidence for the potential impact of these particles and more numerous nanoscale fragments on ice formation in mixed-phase clouds (e.g. Möhler et al.,2007; Pratt et al. 2009; Kanitz et al.,2011; O'Sullivan et al.,2015). Also, recent studies indicate a missing source of INPs beyond mineral dust, with biological particles from terrestrial environments being a likely candidate for initiating freezing in shallow mixed-phase clouds (O'Sullivan et al., 2018). One of the potential sources for these terrestrial INPs are agricultural areas which may contribute between 7 and 75 % to the regional dust burden (Ginoux et al., 2012) due to emissions driven by wind erosion and land management activities such as tilling and harvesting (Hoffmann et al, 2008; Funk et al., 2008; Iturri et al., 2017). Vegetated areas are another source for complex organic aerosol particles associated with leaf detritus (Coz et al., 2010).

One of the characteristics of biological INPs is that they include a vast variety of different particle types, ranging from primary biological particles such as bacteria, fungi and pollen to complex organic particles carrying different ice-nucleating agents and originating from biogenic sources (Schnell and Vali, 1973; Hoose and Möhler, 2012; Murray et al., 2012; Augustin et al., 2013; O'Sullivan et al., 2014; Tobo et al., 2014; Conen et al., 2016; Steinke et al., 2016). An example for complex organic particles are agricultural soil dust particles where the observed high ice nucleation efficiency can be linked to microbiological activity and the presence of organic macromolecules (O'Sullivan et al., 2014; Tobo et al., 2014; Hill et al., 2016; Steinke et al., 2016; Suski et al., 2018). The expression of bacterial and fungal ice-active proteins is highly variable, also because environmental stress (e.g. a change in temperature) can change the structure of ice-nucleating proteins, resulting in a loss of functionality (Pummer et al., 2012). In contrast, some of the organic macromolecules found in agricultural soils are very inert as they are able to withstand physical and chemical treatments, e.g. with heat or exposure to enzymes (Hill et al., 2016). With decaying plant material being one of the sources of these macromolecules (Hill et al., 2016), the need arises to better characterize the ice nucleation properties of plant-derived particles as well as their individual organic components.

Lignin and polysaccharides are integral components of plant cell structures and contribute up to 50 % to plant debris (Williams and Gray, 1974). Proteinaceous components of leaf litter (e.g. enzymes, storage proteins or

structure proteins) vary considerably but have been found to account for up to 15 % (Williams and Gray, 1974). Lipids contribute up to 10 % to dry leaf mass (Graça et al., 2005). Note that only 50 % of the organic matter is accessible through chemical degradative techniques which inadvertently impact the structure of the extracted organic matter (Kögel-Knabner, 2002).

In this study, we investigate the immersion freezing properties of commercially available plant-derived organic compounds such as lignin, polysaccharides, plant wax and plant proteins – which are the main components of decaying plant material – as well as ambient bulk samples rich in plant material. We used commercially available organic compounds as analogues for plant-derived organics. Note that many of the extraction methods for organic matter may cause significant changes in the physicochemical properties of the extracted organic compounds (Kögel-Knabner, 2002). Experiments were conducted at the Aerosol Interactions and Dynamics in the Atmosphere (AIDA) cloud chamber and complemented by drop freezing assay studies using the Ice Nucleation Spectrometer of the Karlsruhe Institute of Technology (INSEKT). From our experimental results we derived temperature dependent parameterizations based on the ice nucleation active surface site (INAS) densities concept (Connolly et al., 2009; Niemand et al., 2012). These parameterizations were then used to estimate upper limits for ambient INP concentrations for complex organic aerosols from vegetated environments.

## 2 Samples and methods

### 2.1 Samples

In Table 1 we describe the samples used in this study, which include commercially available plant-derived organic compounds as well as bulk samples from vegetated environments.

Note that the agricultural dust from harvesting machines (bulk sample) contains roughly 90 % of biological material, e.g. partially intact plant cells and similar particles (Fig. S1). The soil dust sample from Wyoming has been investigated in a recent study by Tobo et al. (2014) finding that organics contribute signficantly to the ice nucleation efficiency observed for size-selected particles (d = 600 nm). Representative microscopy images of all other samples used in this study are shown in the supplement (Fig. S2).

### 2.2 AIDA immersion freezing experiments

Immersion freezing initiated by plant-related particles was investigated in the AIDA cloud chamber (Karlsruhe Institute of Technology, Germany). The AIDA cloud chamber consists of a cylindrical aluminium vessel (volume 84 $m^3$) which is enclosed by a thermally insulated box. The ascent of cloud parcels is simulated by lowering the pressure from ambient levels (about 1000 hPa) to around 800 mbar, and by that lowering the temperature and increasing the relative humidity in the expanding air of the chamber volume.

A fan at the bottom of the AIDA chamber ensures homogeneous mixing (also with regard to temperature and humidity) across the whole chamber volume, except for transition zones near the chamber walls. The overall uncertainty of the mean gas temperature is about $\Delta T = \pm 0.3$ K (Möhler et al., 2006). The absolute water vapor partial pressure is measured with a tunable diode laser instrument and converted into humidity values by leveraging the saturation pressure formulation given in the review by Murphy and Koop (2005). The relative humidity values can be measured with an accuracy of $\Delta RH_{ice} = \pm 5$ % (Fahey et al., 2014).

Particle background concentrations within the cloud chamber are typically below 0.1 $cm^{-3}$. For the immersion freezing experiments presented in this work, aerosol samples were injected into the cloud chamber by using a rotating brush generator (RBG-1000, Palas GmbH) for dry dispersion. Additionally, impactor stages were used

to eliminate particles larger than 3 to 5 µm. The aerosol size distribution at the beginning of each experimental run was measured by combining data from an Aerodynamic Particle Sizer (APS, TSI, Model 3321) and a Scanning Mobility Particle Sizer (SMPS, TSI, Model 3076). The combined aerosol size distributions are used to estimate the available aerosol surface area based on volume-equivalent sphere diameters which then results in an estimate of the geometric surface area.

Upon reaching water saturation during an expansion experiment, aerosol particles within the cloud chamber are activated to droplets and may freeze subsequently. Ice crystal number concentrations are measured with two optical particle counters (WhitE-Light Aerosol Spectrometer, welas1 and welas2, series 2300 and 2500, PALAS GmbH) with size ranges of 0.7 – 46 and 5 – 240 µm in optical particle diameter, respectively (Wagner and Möhler, 2013). Ice crystals are discriminated from droplets by choosing a size threshold which is evaluated individually for each experiment.

**2.2 Droplet freezing assay studies**

To investigate the freezing of suspensions created with the bulk samples and hence to account for freezing caused by particles larger than 5 µm, a droplet freezing technique was employed. The Ice Nucleation Spectrometer of the Karlsruhe Institute of Technology (INSEKT) setup (Schiebel, 2017) is based on the droplet freezing assay originally developed at Colorado State University (Hill et al., 2014).

Suspensions were created from bulk samples, combining 2 mg of material with 20 ml of deionized water (resistivity about 18 MΩ) which has been passed through a filter with a pore diameter of 0.1 µm (Whatman Puradisc 25). Suspensions were shaken by hand (about 1 min) and the suspension tube was then submerged in an ultrasonic bath (5 min) to promote dispersion of the particles. In addition to the original suspensions, we also created suspensions with a dilution factor of 15 and 225 by adding filtered deionized water in proportion. Original and diluted suspensions were partitioned into 192 wells (aliquot volume: 50 µL) of a sterile polypropylene polymerase chain reaction (PCR) tray, with 32 wells set aside for blank measurements, i.e. freezing of particle-free filtered deionized water. These blank measurements are used for determining the background which is then subtracted from the observed freezing curves. In this study, droplet freezing was measured at a cooling rate of 0.33 K/min. Cooling is achieved by flowing chilled ethanol through a custom-made aluminium block which encloses the bottom part of the PCR tray. The overall temperature uncertainty is $\Delta T = \pm 0.3$ K (Schiebel, 2017).

**2.3 Ice nucleation active surface site densities**

For all experiments, the ice nucleation efficiency was quantified by calculating the ice nucleation active surface site (INAS) density $n_s$. The $n_s$ values were derived by scaling the observed ice crystal number concentration $n_{ice}$ with the available aerosol surface $A_{aer}$ (Connolly et al., 2009; Niemand et al., 2012). Exemplary size distributions for leaf litter and lignin are shown in S3.

For the cloud chamber experiments, the aerosol surface $A_{aer}$ [µm$^2$/cm$^3$] was calculated from the APS and SMPS size distribution data using volume-equivalent sphere diameters (Möhler et al., 2006). In this study, it was assumed that all aerosol particles are activated to droplets upon reaching water saturation. Hence, the full aerosol surface area was considered to be available for immersion freezing. The ice crystal number concentration $n_{ice}$ was derived from particle size distributions measured with the optical particle counters welas1 and welas2, in conjunction with a size threshold above which particles are counted as ice crystals. Based on the measurement

uncertainties of the observed ice crystal concentration $\Delta n_{ice}/n_{ice} = 0.2$ and the aerosol surface area concentration $\Delta A_{aer}/A_{aer} = 0.35$, the resulting uncertainty of the INAS density is $\Delta n_s/n_s = 0.4$ (Ullrich et al., 2017).

For the droplet freezing studies, the INAS density values were derived from normalizing the cumulative INP concentration $n_{ice}$ with the specific aerosol surface $A_{aer}$ [$m^2/g$] derived from Brunauer-Emmett-Teller (BET) surface measurements. For our INAS density uncertainty analysis, we considered only the uncertainty of the cumulative INP concentrations which is based on statistics. Confidence intervals (at 95 %) have been estimated according to the improved Wald interval which implicitly assumes a normal approximation for binomially distributed measurement errors (Agresti and Coull, 1998). Hence, in our INAS density analysis, we neglected the uncertainties of the BET surface measurements which are in most cases considerably smaller (i.e. $\Delta A_{aer}/A_{aer} <$ 0.1) than the previously described statistical uncertainties of the cumulative INP concentrations (Hiranuma et al., 2015a). Another source of uncertainty – which is considerably more difficult to quantify – was the contribution of larger particles. These larger particles may sediment quickly within the suspension and were probably under-represented in the sampled aliquots. Thus, the particle surface area available for freezing was most likely overestimated in some cases. However, to fully understand this effect more studies are needed. Additionally, suspending particles in water may lead to the desorption and potential redistribution of soluble material. This change in soluble material could also lead to differences in the observed ice nucleation properties when comparing cloud chamber experiments with droplet freezing studies.

**3 Results and discussion**

In Fig. 1 we present results from AIDA cloud chamber experiments with commercially available plant-related organic compounds and natural samples (see Table 1). For comparison, we show the ice nucleation activity of microcrystalline cellulose (Hiranuma et al., 2015b), which is a prevalent natural polymer deriving from plant fragments, leaf litter, wood fiber, non-wood fiber and/or even microbes (Quiroz-Castañeda and Folch-Mallol, 2013; Vlachou et al., 2018). We also show the ice nucleation efficiency of agricultural soil dusts investigated in a study by Steinke et al. (2016) as well as an estimate for leaf litter from a study by Schnell et al. (1972). The ice nucleation activity of each sample is expressed as the INAS density $n_s$.

Figure 1 shows that the observed ice nucleation efficiencies of most individual plant-related organic compounds tend to be lower in comparison to samples from natural environments. However, there is a large spread in INAS density values when comparing between different plant-related organic compounds. Particularly noticeable is the low ice nucleation efficiency observed for plant protein for which freezing was observed only below 248 K. In this study, we tested two different types of plant proteins (PROT_R, PROT_SOY), derived from soy or rice (not differentiated in Fig. 1). Only lignin (LIG) shows an ice nucleation activity as low as the plant protein samples. Alginate, pectin, and starch (which mainly consist of highly complex polysaccharides) are similarly ice-active as microcrystalline cellulose (Hiranuma et al., 2015b) and desert dusts (Ullrich et al., 2017 – not shown in Fig. 2). Above 250 K, the complex polysaccharides investigated in this study (ALG, PEC, STAR_P, STAR_C) tend to be more ice-active than cellulose. Our data also indicates that the temperature dependence of the polysaccharides investigated in this study is possibly less pronounced than for cellulose. Note that this finding is based only on a few data points due to the low observed ice nucleation efficiency above 252 K.

Of all plant-related compounds, carnauba wax (LIP) shows the highest ice nucleation efficiency, comparable to decaying leaves and two agricultural samples, i.e. dust from a sugar beet field (AGDUST_WYO) and material collected from harvesting machines (AGDUST_HARV). Carnauba wax is a mixture of hydrocarbons, aliphatic

esters and fatty alcohols (Vandenburg and Wilder, 1970) with an average chain length of 50 carbon atoms (Basson and Reynhardt, 1988). Crystalline fatty alcohols (C16 - C18) have been highlighted recently in a study by DeMott et al. (2018) with regard to their ability to nucleate ice at 261 K via condensation freezing. Based on theoretical considerations, hydrocarbons with long chains are potentially very good at initiating ice formation
(Qiu et al., 2017) but conclusive experimental evidence is still missing. Hence, these theoreretical considerations might provide an explanation for the high ice nucleation ability of carnauba wax.

For samples like the agricultural soil dusts and the leaf litter investigated in this study, some studies (e.g Schnell and Vali,1973; Steinke et al.,2016) have found similarly high ice nucleation efficiencies.
In contrast, at 258 K, leaf litter from the Arctic consisting of birch and grass leaves (Conen et al., 2016) has been
observed to show relatively low ice nucleation efficiencies compared to leaf litter in our study based on AIDA results and similar efficiencies when comparing against our droplet freezing assay.
Hence, the high INAS density values observed in our cloud chamber studies can be interpreted as upper limits for the ice nucleation efficiency of ambient plant-related aerosol particles. Note that for our leaf litter samples we did not differentiate between samples collected at different points in time and for different species. Due to the
high variability it was not possible to clearly derive a seasonal trend from the observed ice nucleation efficiencies.

In Fig. 2, we show INSEKT-derived INAS density values for selected samples investigated in the previously described AIDA cloud chamber studies. For every sample at least two experimental runs were conducted, using freshly prepared suspensions for each run. The PROT_S sample was investigated to establish the lower boundary
of ice nucleation activity observed for plant components whereas the AGDUST_HARV and the LEAF samples were used to represent ambient samples. Note that for the droplet freezing experiments, the INAS densities are evaluated based on the specific surface areas derived from BET measurements rather than the geometric surface areas which were used for analyzing the AIDA experiments. The droplet freezing experiments are complementary to the cloud chamber studies as they deliver insights regarding the freezing properties of the bulk
material, in particular with regard to including particles larger than 5 μm which are largely eliminated by impactor stages in our AIDA experiments. Also, observing the freezing of bulk suspensions allows for quantifying the immersion freezing efficiencies at a lower supercooling which are more difficult to quantify in AIDA cloud chamber studies. For leaf litter we observe that INAS density values agree well between INSEKT and AIDA experiments. Similarly for plant protein (PROT_S), the agreement is reasonably. For
AGDUST_HARV, there is a difference of approximately more than one order of magnitude which is possibly caused by larger particles being undersampled due to sedimentation within the suspensions.

Figure 2 shows that the hierarchy in ice nucleation activities is similar as observed in the AIDA cloud chamber experiments, with leaf litter and agricultural dust being the most ice-active samples. The steep onset of ice nucleation observed for the agricultural dust at 267 K suggests a contribution from biological particles (Suski et
al., 2018). In contrast, the reasons for the steep onset observed for the leaf litter sample are a bit more unclear as most studies investigating primary biological particles have observed freezing onsets and high ice nucleation efficiency already at temperatures above 260 K (see references in Hoose and Möhler, 2012). However, one recent study has found indications for macromolecules associated with microbial activity being ice-active at about 258 K (O'Sullivan et al., 2015). Soy protein particles initiate ice formation at higher temperatures (i.e.
already below 258 K) than observed in AIDA cloud chamber experiments, but the overall ice nucleation efficiency is still lower than for the complex organic samples from natural environments. Unfortunately, it was

not possible to reliably determine INAS density values for carnauba wax (LIP) due to its very low dispersibility. Figure 2 also shows the INAS density values observed for illite as a proxy for freezing induced by mineral dust.

In conclusion, the results from the droplet freezing studies confirm the trend observed in our AIDA cloud chamber experiments, with particles from vegetated and agricultural environments being highly ice-active, whereas individual organic compounds tend to be lower in their ice nucleation efficiencies. It should be noted that the organic compounds investigated in this study may not fully represent the complexity of real organic compounds in plants which often include mixtures, e.g. ligno-polysaccharide complexes with unknown chemical structures (Kögel-Knabner, 2002). At temperatures above 260 K, the gap between individual plant-related compounds and particles from natural environments may be attributed to primary biological particles (e.g. fungi and bacteria) according to our droplet freezing measurements of harvesting dust. For example, ice nucleation efficiencies observed for particles generated from leaf litter fall within the lower range of values observed for bacteria (Hoose and Möhler, 2012).

There are, however, also differences between the ice nucleation efficiencies derived from AIDA cloud chamber experiments and droplet freezing studies, which strongly dependent on the aerosol type. Some of these differences might be explained by differences in the evaluation of the INAS density values which are either related to the geometric surface or the specific surface area. For illite, normalizing by BET surface area results in INAS density values which are one order of magnitude lower compared to values derived by using geometric surface estimates (Hiranuma et al., 2015). Also, for some samples there are possibly differences in the effective size distribution due to agglomeration or low dispersibility in the suspensions. In contrast, the dry dispersion method (i.e. the rotating brush generator) is more likely to encourage disaggregation of particle agglomerates. Similar differences regarding the freezing of aqueous suspensions in comparison to dry dispersion experiments have been observed in other studies as well (Hiranuma et al., 2015a; Hiranuma et al., 2019).

Our experimental results suggest that the main components of decaying plant material (i.e. cellulose and lignin) are not very good predictors of ice nucleation by ambient plant-related particles. However, the INAS density values observed for leaf litter and agricultural dust may help to constrain the upper limits of their respective ambient INP concentrations. The INAS density values for leaf litter and agricultural dust can be described by temperature-dependent functions, with

$$n_{s,leaf} = \exp(-0.246 \cdot T_{leaf} + 84.681) \qquad\qquad r^2 = 0.70 \qquad\qquad (1)$$

and

$$n_{s,agri} = \exp(-0.541 \cdot T_{agri} + 157.471) \qquad\qquad r^2 = 0.84 \qquad\qquad (2)$$

Note that these functions are only valid within certain temperature ranges, i.e. $T_{leaf}$ = [243, 258] and $T_{agri}$ = [245, 255], with all temperatures given in [K]. Equations 1 and 2 have been derived from the cloud chamber experiments exclusively and are represented in Fig. 2. Note that based on our droplet freezing experiments, both of these aerosol types may have relatively sharp ice nucleation onsets at 257 K (leaf litter) and 267 K (agricultural dusts).

Figure 3 shows a comparison between ambient INP concentration derived from precipitation samples from several sites in the United States and Europe (Petters and Wright, 2015) and estimates for INP concentrations from leaf litter (eq.1) and agricultural dust (eq.2). Note that ambient INP measurements may scatter significantly

275 more than found in the study by Petters and Wright (2015), with deviations of up to four orders of magnitude between different studies (Kanji et al., 2017).

Ground-based measurements for leaf litter concentrations range between 30 ng/m$^3$ to 1 µg/m$^3$ (Hildemann et al., 1996; Sánchez-Ochoa et al., 2007). Sánchez-Ochoa et al (2007) use cellulose found in aerosol particles as a proxy for plant debris concentrations, relying on observations at 6 European sites for a time span of two years,

280 and with two of the sites being located on mountains. Hildemann et al. (1996) used higher alkanes (e.g. ocurring in plant waxes) to fingerprint plant debris in aerosol particles sampled in the greater Los Angeles Area. For agricultural dust, ground-based concentration vary between <10 and 100 µg/m$^3$, with up to 800 µg/m$^3$ observed occasionally for very strong wind erosion events (Gillette et al., 1978; Sharratt et al., 2007; Hoffmann and Funk, 2015). Annually averaged boundary layer concentrations for desert dust vary between 0.1 and 30 µg/m$^3$ (Ginoux

285 et al., 2001) which is comparable to the aforementioned concentrations of complex organic particles. Anthropogenic dust sources contribute roughly 25 % to the global dust burden, with regional variations ranging from 7 to 75 % (Ginoux et al., 2012). In areas with intense agricultural land use, e.g. in eastern North America, India, eastern China, and Europe, anthropogenic dust emissions contribute generally more than 60 % to the total dust burden (Huang et al., 2015). Note, however, that there is a substantial uncertainty regarding the number and

290 size of particles emitted from agricultural as well as their transport to cloud altitudes and the resulting atmospheric lifetime. This uncertainty is rooted in a lack of emission flux data above 5-10 m which is the height at which dust fluxes from agricultural areas are commonly observed, e.g. in the study by Zobeck and Van Pelt (2006). Using eqs. 1 and 2 and assuming an aerosol surface area of 1 and 36 m$^2$/g as measured by BET analysis, we can derive order-of-magnitude estimates for the expected atmospheric INP contribution from leaf litter and

295 agricultural dust. In Fig. 3, we have scaled down agricultural dust INPs by a factor 100 and leaf litter INPs by a factor of 10 to at least partially account for transport losses. Scaling factors are derived from model results presented in Hoose et al. (2010), using vertical profiles for desert dusts and biological particles as rough proxies for the samples investigated in this study.

The estimates presented in this study should be considered as upper limits, with emission fluxes of organic

300 particles acting as INPs being poorly constrained and more detailed modelling case studies needed. We find that plant-derived organic INPs from leaf litter and agricultural areas are within the same order magnitude as INP concentrations derived from precipitation measurements and field campaigns (Petters and Wright, 2015; Kanji et al., 2017). This finding further emphasizes the potential of plant-related sources to contribute to ambient INPs.

**Section 4: Conclusions**

305 Complex organic particles are emitted from terrestrial sources, with wind erosion, soil cultivation and harvesting crops as potential main drivers for emissions of organic matter associated with plant debris and decomposed residues (Funk et al., 2008; Hoffmann et al., 2008; Coz et al., 2010; Ginoux et al., 2012). These sources are becoming increasingly important in the global view, as climate change, soil degradation and excessive land use will promote dust emissions from agriculturally used areas. In this study, we investigated the immersion freezing

310 properties of plant-related organic particles and samples from vegetated environments. We used a combination of AIDA cloud chamber and INSEKT droplet freezing experiments to cover a temperature range between 242 and 267 K. Our experiments show that the samples with a complex organic composition are equally or more ice-active than individual plant-related compounds. Lignin and plant protein samples are inefficient INPs, whereas starches, alginate and pectin show moderate to high ice nucleation efficiencies. Surprisingly, carnauba wax –

315 which is a mixture of aliphatic esters and fatty acids – shows the highest ice nucleation activity of all organic

compounds investigated in this study. INP estimates based on our cloud chamber experiments lend themselves to the hypothesis that aerosolized particles from leaf litter and agricultural areas are potentially important contributors to atmospheric INPs. However, the high ice nucleation efficiency of these particles could not be fully explained by the ice nucleation activity of individual organic compounds commonly found in plant tissue,

potentially indicating a contribution from primary biological particles or organics associated with microbial activity. Thus, further future studies are indeed demanded and warranted.


**Author contributions**

IS and NH designed and conducted the experiments, with contributions from KH, OM and NSU. PGW conducted the BET surface measurements and NT provided the SEM images. IS and NH analyzed the data. IS

prepared the manuscript with input from all co-authors.

**Acknowledgements**

This study was conducted with financial support from the Carl-Zeiss-Foundation and the German Science Foundation (DFG) through the research unit INUIT (FOR 1525, MO668/4-2). Dr. Steinke was also funded in part by the U.S. Department of Energy (BER), through the Early Career Program. Dr. Umo was funded by the

Alexander von Humboldt Foundation Germany under the grant agreement No. 1188375. Some microscopy research and sample pre-characterization were preformed in the Environmental Molecular Science Laboratory (under User Proposal 49077), a DOE Office of Science User Facility sponsored by the Office of Biological and Environmental Research and located at Pacific Northwest National Laboratory.

Hinrich Grothe (Vienna University of Technology) is acknowledged for having provided several organic

samples (lignin, carnauba wax, pectin, alginate, starches) investigated in this study.

The IMK-AAF technicians team (Georg Scheurig, Rainer Buschbacher, Tomasz Chudy, Olga Dombrowski and Steffen Vogt) is acknowledged for their continued support in ensuring a smooth operation of the AIDA cloud chamber.

**Competing interests**

The authors declare no competing financial interests.

**Data management**

All data in this manuscript will be made available as part of a KITopen data repository.

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

**Tables**

| Sample name | Acronym | Sample preparation/manufacturer |
|---|---|---|
| *Ambient bulk samples dominated by decaying plant material* | | |
| Leaf litter | LEAF | Dry leaf debris from either spruce or maple trees in Southwestern Germany, dried at 313 K, milled and sieved for particles smaller than 150 µm (collected in spring and autumn in the years 2014, 2015, and 2016) |
| Agricultural dust | AGDUST_HARV | Dry plant material collected from filters of harvesting machines after rye and wheat harvests in Northwestern Germany, sieved for particles smaller than 63 µm (collected in summer 2016) |
| Agricultural soil dust | AGDUST_WYO | Top soil samples collected in Wyoming on sugar beet fields (collected in spring 2011) |
| Alginate | ALG | C.E. Roeper GmbH (article no. NA 4012) |
| Lignin | LIG | Sigma-Aldrich (article no. 370959 and 471003) |
| Lipids (Carnaubawax) | LIP | Sigma-Aldrich (article no. 243213) |
| *Plant-related organic compounds* | | |
| Pectin | PEC | Herbstreith & Fox KG (article no. AU 015 H I) |
| Protein (Rice, soy) | PROT_R, PROT_S | Erdschwalbe (article no. 30676 and 30744, food grade quality) |
| Starch (Potato) | STAR_P | Mueller's Muehle GmbH (food grade quality) |
| Starch (Corn) | STAR_C | Unilever (food grade quality) |

**Table 1:** Overview of samples used for ice nucleation experiments.

**Figures**

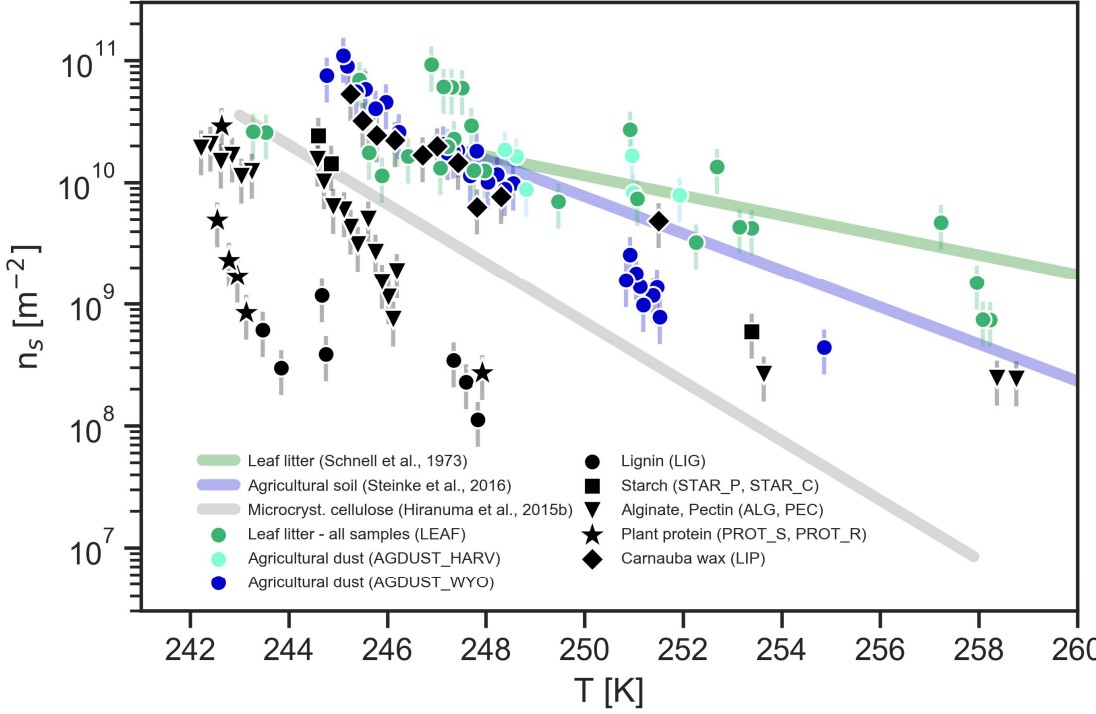

**Figure 1:** Immersion freezing results for plant-related organic compounds compared to ambient samples – ice nucleation efficiency expressed as INAS density values based on AIDA cloud chamber experiments.

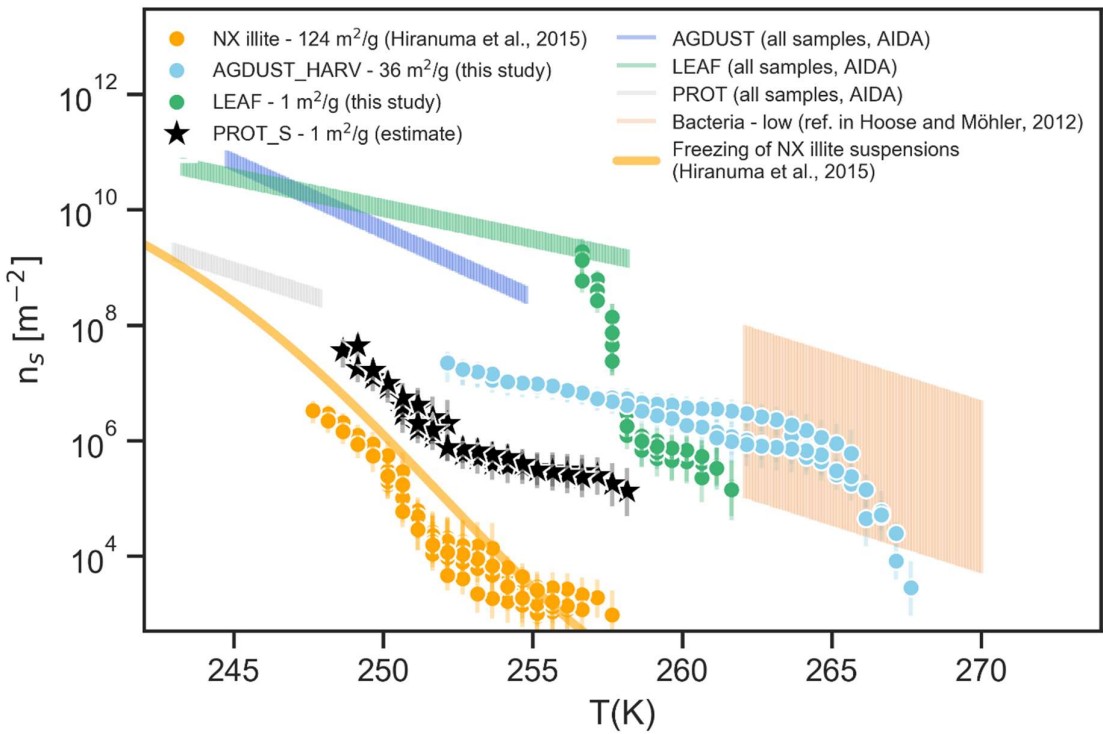

**Figure 2:** Immersion freezing results for selected plant-related samples and illite, comparing INSEKT-derived INAS density values to results from AIDA experiments (Fig.1) – ice nucleation efficiency expressed as INAS density values based on INSEKT droplet freezing experiments and specific surface areas indicated in the legend.

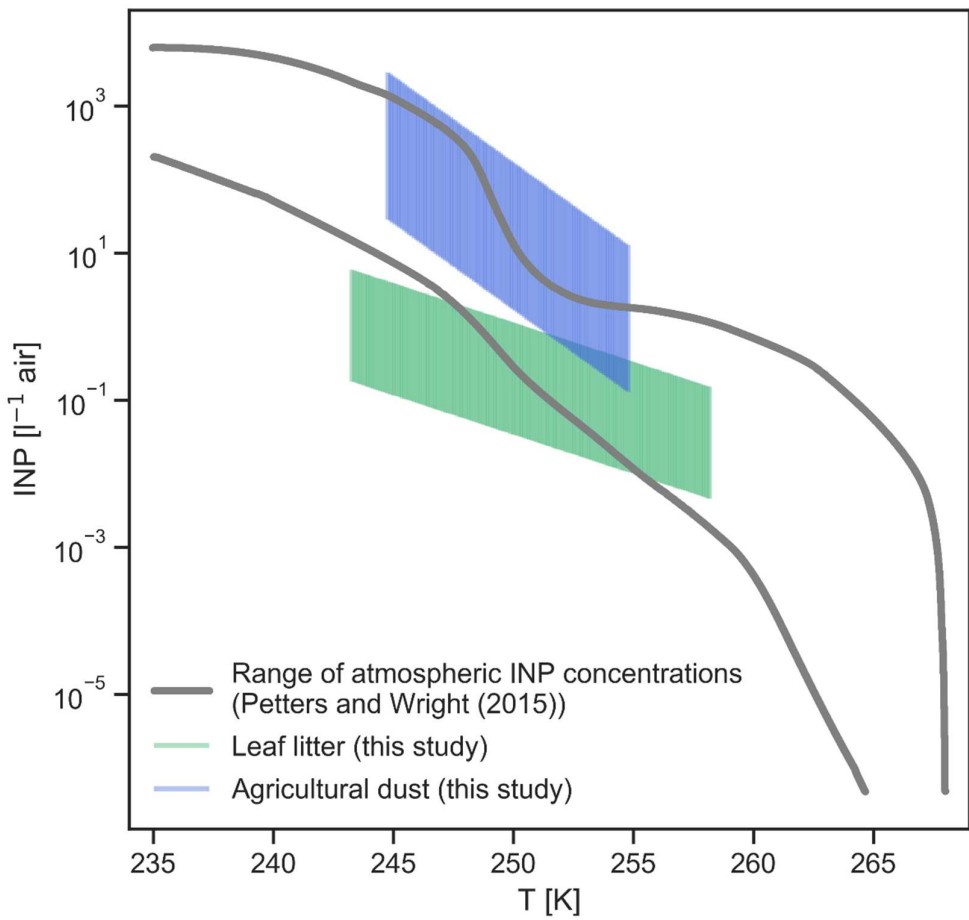

**Figure 3:** Comparison between atmospheric INP concentrations (Petters and Wright (2015)) and estimates for INPs from leaf litter and agricultural dust based on AIDA cloud chamber experiments.