# Peer review of "Complex plant-derived organic aerosol as ice-nucleating particles – more than a sum of their parts?"

_Atmospheric Chemistry and Physics, 2019_

## Referee Comment (RC1) · Anonymous Referee #1 · 13 Dec 2019

The manuscript uses two different techniques to examine INP number concentrations for a number of plant and soil derived samples, namely an expansion chamber, AIDA, and a cold stage method, INSEKT. Measurements were certainly conducted under the necessary care, and the results as such are interesting. However, unfortunately, the impression arises that the text was written in a hurry, still trying to give the impression that the data are really important, and in this course, overlooking topics that should have been discussed in much more detail if claims as those made in the text are to be justified.

My main criticisms are: a) the effect of micro-organism as an INP is largely not dis-

cussed in the interpretation of the retrieved results, although for leaf litter and the agricultural samples, it cannot be excluded that this plays a role; b) the amount of plant derived INP in the atmosphere is derived without properly motivating all the parameters used in the calculation, which, however, will largely influence the atmospheric importance of these respective INP.

More specifically, it has to be said that throughout the text, there is a somewhat strange shortage of mentions of biogenic INP, which, based on their nature, have to be connected to the herein examined plant material samples. In the absence of any physical or chemical test that could have altered the respective samples (e.g., heating, treatment with H2O2), there is no information about the nature of the INPs, therefore any observed activity in the agricultural sample and the leaf litter may originate from the sample material itself or from micro-organisms connected to it (bacteria, fungi, lichen). The overall text has to be revised in connection to this issue.

Also, for all of those samples from plant-related organic compounds, the sources for the cited quantities for agricultural dust and leaf litter have to be explained in more detail. These numbers are used for a very general calculation afterwards, but the reader needs to know if the given concentrations are valid only locally or for a larger area or worldwide and in which altitudes. A mentioning of "occasionally for very strong wind events" (line 256) rather gives the impression that these values are of no general use in this context, and more information is needed. Also some mentioning of the processes that make particles from dust and leaf litter airborne should be added.

These issues and others to be found in my comments below makes a thorough revision of the text necessary before it can be accepted for publication in ACPD.

Specific comments:

line 46: "biological aerosol particles" come a bit out of nowhere, here, and it would be good to shortly mention first, that these, too, can be efficient INP (as done for the mineral dust in the previous sentence).

[Figure]

line 66: Your sentence "decaying plant material being one of the major sources of these macromolecules (Hill et al., 2016)," gives a somewhat wrong impression, as (from what I understood from the literature you cite here) these macromolecules may also originate from biological entities. In their abstract, it says: "Organic INPs . . . may originate from decomposing plant material, microbial biomass, and/or the humin component of the SOM."

line 114: It should be added that the identification of an hydrometor as ice is simply based on its size.

line 167: These individual plant-related organic compounds likely miss the contribution of micro-organisms, so it is not so astounding that they show a comparably low ice activity ("in comparison to samples from natural environments"). A discussion on this is missing completely from the text.

line 175: There is only a VERY low number of points for these "complex polysac-carides" at these higher temperatures (one, one, three, for three different ones of these samples (where the one with the three points seems to consist of two different materials, ALG and PEC)). - Particularly the three dots for ALG/PEC seem to be rather a background, as there is no change in the signal over the temperature range from 254 to 259 K, making this really look like a background issue. Interpreting this as a "weaker dependence on temperature" over-interprets what can be learned from these few data-points.

line 188 ff: The abstract of Conen et al. (2016) says: "Together, both findings suggest that decaying leaves are a strong emission source of IN to the Arctic boundary layer." And they examined litter consisting "of entire leaves and large fragments thereof, mainly from Betula nana and various grasses." – I wonder how this fits to what you wrote: ". . ., leaf litter from the Arctic consisting mainly of grass leaves has been observed to show relatively low ice nucleation efficiencies (Conen et al., 2016), . . ." . Neither seems grass to be the major component of the litter examined by Conen et al. (2016) (or

did I overlook something in the Conen-study?), nor do they claim that this is a low ice nucleation efficiency. Temperature wise, data in Conen et al. (2016) only go down to -15°C, while the temperature range into which your results fall are generally below that, which has to be acknowledged when comparing values. Please revise this part of the text.

line 208-209: Is there any explanation why for the two samples discussed above large particles were not a problem and then why they might be a problem here? How else does this one sample differ from these two samples above?

212-213: Why should there not also be a contribution from micro-organisms (= biological particles) in the leaf litter sample? After all, originally it was leaves from which Pseudomonas syringae was derived (Maki 1974). While the temperature of the signal's increase does not suggest the influence of a bacteria such as P. syringae, the presence of other micro-organisms could still be responsible for this increase. Testing with e.g., heat or acids or other methods could have increased the understanding.

line 248/249 and line 254 ff: The explanation on how INP concentrations were obtained, i.e., the atmospheric concentrations assumed for leaf litter and agricultural soil dust, needs to be given first, before referring to the derived INP concentrations shown in Fig. 3. Also, it should be discussed in more detail that these indeed are upper boundaries (you mention "very strong wind erosion events"). And also, please motivate the aerosol surface area (1 m2/g). This would need to be the surface area ascribed to leaf litter or to the dust, exclusively. Is this a reasonable value? This needs to be discussed. Also: the mass concentrations: are they to be expected over larger areas, or were these just measured near sources. This is important if you want to make statements on atmospheric wide INP concentrations and the importance of leaf litter and agricultural soil dust in this context.

line 265: Could it be that this sentence is based on using unrealistically high atmospheric concentrations of materials that were examined and then using this as an argument for the importance of the present study? For enabling your readers to judge that, motivate the 1 m2/g that you used above, and the mass concentrations of leaf litter and dust, as said above.

line 267: The "Section 4: Conclusion" comes a bit abrupt and contains statements that should either be given in a discussion part or otherwise earlier – giving new points of discussion (as the different sources that may emit organic particles or the global extent of the areas that may contribute) is not something that should appear in the conclusions for the first time.

line 277: Why were INSEKT data not also used to obtain atmospheric INP concentrations? As you say here, the measurements with this instrument were done to get results on a wider temperature range, but then they are not used in the further evaluation. Maybe it could also be seen if AIDA or INSEKT give the more trustworthy data? At least this could be discussed when comparing to the data from literature.

line 282-284: This sentence might have to be revised if the used values for mass concentration and aerosol surface area are revised or cannot be well justified.

line 284-286: Here it is again not clear why macromolecules from micro-organisms were not considered, as they provide an obvious explanation.

Figure 3: As also already mentioned above, I wonder if you want to imply here that the atmospheric INP mostly come from leaf litter and agricultural dust? There is a lot of literature around that ascribes ice activity at temperatures of roughly < -20°C to desert dust, so I wonder if you really want to challenge this. If you feel your data is strong enough to do that, come up with a good justification. If you don't trust your derived concentrations so much, make it clearer in the text that you present the absolute possible maximum and that likely values from the lower end of the ranges you give or even blow are more likely.

Technical comments:

[Figure]

line 51-52: Check the style of the citations (no brackets should be used in a bracket) Table 1: Somehow a "555" shows up at the right side of the table. Probably a line number gone wild? Figure 3: "." is missing at the end of the caption.

Literature:

Conen, F., E. Stopelli, and L. Zimmermann (2016), Clues that decaying leaves enrich Arctic air with ice nucleating particles, Atmos. Environ., 129, 91-94, doi:10.1016/j.atmosenv.2016.01.027.

Hill, T. C. J., P. J. DeMott, Y. Tobo, J. Froehlich-Nowoisky, B. F. Moffett, G. D. Franc, and S. M. Kreidenweis (2016), Sources of organic ice nucleating particles in soils, Atmos. Chem. Phys., 16(11), 7195-7211, doi:10.5194/acp-16-7195-2016. Maki, L. R., E. L. Galyan, M.-M. Changchi, and D. R. Caldwell (1974), Ice nucleation induced by Pseudomonas-Syringae, Appl. Microbiol., 28(3), 456-459.

---

## Referee Comment (RC2) · Anonymous Referee #2 · 20 Jan 2020

The authors present an interesting study aimed at isolating the ice nucleating (immersion freezing) active components of vegetation (e.g., leaf litter and harvesting debris). Particles were generated from commercially-available compounds, such as lignin and carnauba wax, and ice nucleating number concentrations measured as a function of temperature. For comparison to prior work and to normalize the findings, ice nucleating activity was converted to a per-unit-surface-area basis (ice nucleation active surface site density, INAS). To cover the full range of temperatures, two methods – a microdroplet assay and an expansion cloud chamber – were applied.

The idea of a systematic study to develop more basic understanding of how complex

atmospheric particles behave in clouds is commendable, and this group is well known for their expertise and impressive laboratory capabilities in probing ice nucleating properties of aerosols. Nevertheless, this study has some important gaps in the presented work that limit the conclusions that can be drawn and the applicability of this work toward improving understanding. I recommend major revision before publishing.

First, ice nucleating active (INA) bacteria were identified decades ago and it is well known that vegetation and leaf litter – depending on type – can host dense populations of these bacteria. This component is discussed in the introductory materials, but not brought up again in comparison with the results for vegetation samples. Why not study P. syringae (as a model for this component) with the same systems and compare to that? Further, the INA component in bacteria is a lipoglycoprotein (with a particular structure that enables its activity), which presumably inspired some of the choices in Table 1. But this is not explained; and in any case, this is also already well known, so it is not clear what was to be accomplished through the selections made for study unless it is implied that other proteins, lipids, etc. might have IN activity as well (if so, why?). The idea of other "unknown" organic constituents being important (e.g., the macromolecules proposed in earlier work by other groups) is certainly raised, but is not explicitly investigated here – except perhaps by ruling out activity from larger particles composed of the selected compounds.

Second, the argument is made that using commercially-available components is preferable because "many of the extraction methods for organic matter may cause significant changes in the physicochemical properties of the extracted organic compounds". Why is this not true also for the commercial products? There is no discussion of how these are manufactured, which seems to be important for the proteins in particular if they are to be considered analogs for natural components. I also have questions regarding the process for generating particles of carnauba wax, which was the only component identified as having significant IN activity: on line 216 it is stated that, "Unfortunately, it was not possible to reliably determine INAS density values for carnauba wax (LIP)

due to its very low dispersibility." It is appreciated that generating reproducible particles from solid samples is very difficult, but the uncertainties associated with this should be quantified and carried through the analyses. The results for carnauba wax are noted to be surprising (lines 284-285) but few fully satisfying reasons for this result can be deduced from the present study (some ideas are presented in lines 177-185).

A third major point with regard to atmospheric implications is that while soils, leaf litter, harvest debris, etc. can have high densities of INA bacteria or other ice-active components, the mobilization of particles containing those components into the boundary layer, and further, to altitudes where they can impact cloud formation, is a different matter. Limited prior studies suggest there is no direct relationship between surface concentrations and atmospheric concentrations and the atmospheric concentrations become relevant only under conditions where the surface is strongly disturbed (as alluded to in the text). Thus the implications of any findings with respect to atmospheric processes have to be tempered by this consideration. In particular, the concluding sentence of the Abstract, "In contrast, complex biological particles may exhibit ice nucleation activities which are up to two orders of magnitude higher than observed for cellulose, making ambient plant-derived particles a potentially important contributor to the population of ice-nucleating particles in the troposphere" is not a unique conclusion from this work but has been suggested previously, and needs to be modified to acknowledge that the relationship between the surface and ambient concentrations needs to be better understood before quantifying the importance of this source on regional and global scales.

I have additional comments for consideration, as follows.

It is stated that for some of the tested samples, the AIDA and microdroplet methods agree (lines 206-208). However, there is no overlap between these methods, and the surface area determinations use very different approaches, calling this agreement into question. The particle background concentrations for AIDA are stated (line 107) as 100 L-1. Comparing to Figure 3, I'm unclear how this is taken into account; the x-axis

[Figure]

scales on Figures 1 and 2 are different, indicating that AIDA is limited at the warmer temperatures, presumably due to this background?

Prior work by Hiranuma et al. (2015b) is cited for data on cellulose for comparison to the present work. The intercomparisons published by Hiranuma et al (2019) are also cited, however, in that study, it is noted that "While the diverse instruments employed in this study agree in that cellulose has the capacity to nucleate ice, their quantitative agreement is poor. Unfortunately, it is not possible yet to say what the cause of this disagreement is." Does this statement apply to the two techniques used in the manuscript? Hiranuma et al. (2019) also call for "comprehensive studies on the ice nucleation activity of other important plant structural materials, such as cellulose polymorphs, lignin materials, lipids, carbohydrates and other macromolecule saccharides", so the present study is a nice follow-on to that recommendation. However, the issue of whether follow-on studies are premature at this point, if there are fundamental questions regarding the measurements and their interpretation, needs to be addressed.

Line 89: "ambient samples from vegetated environments": my comments above assume these are bulk samples and not obtained by filtering of ambient air. If my interpretation is correct, perhaps the language here needs to be clarified.

Line 148: Brunauer is misspelled. The uncertainties introduced by the different estimates of surface area should be more thoroughly discussed and represented in the figures (how are the uncertainty bars in the figures computed – is this from the variation in the repeat experiments, or does it include other considerations such as surface area?)

Line 174: Desert dust (Ullrich et al., 2017) is mentioned for comparison, but not shown?

Line 204: Is the background for the microdroplet method shown here or in another publication?

Line 110, 148: could these aerosol size and surface area distributions be shown in the

Supplementary Material? This is potentially useful information for other studies that might seek to explore similar science questions with other techniques.

---

## Author Comment (AC1) · 12 Mar 2020

***Response to Reviewers:*** *Complex plant-derived organic aerosol as ice-nucleating particles – more than a sum of their parts?*

We would like to thank both reviewers for carefully evaluating our manuscript and for providing valuable feedback. In the following, we want to respond to your overarching comments as well to your technical points.

**Response to Reviewer 1**

*General Comment 1: […] My main criticisms are: a) the effect of micro-organism as an INP is largely not discussed in the interpretation of the retrieved results, although for leaf litter and the agricultural samples, it cannot be excluded that this plays a role […]*

In our study, we deliberately focus on organic compounds which can be found in plant tissue, but which are not directly related to microbial activity. Our study should be viewed as a first step towards a better understanding of complex organic particles from biogenic sources, with organic compounds associated with primary biological particles being even more diverse and thus also more variable in their ice nucleation properties. Observed ice nucleation properties of fungi and bacteria vary over roughly six orders of magnitude (see references in Hoose and Moehler (2012)) and recently there have been results questioning the stability of the ice nucleating proteins responsible for the high-temperature ice nucleation efficiency (Polen et al., 2016). Hence, even though we definitely agree with the potential impact of microorganisms, especially at temperatures above 263 K, we have chosen in this study to investigate only a subgroup of the (presumably more stable) organics which play a role in determining the ice nucleation properties of particles derived from leaf litter and soils. These stable organics can be viewed as a lower limit for the ice nucleation activity of more complex particles. We have emphasized the potential role of microbial activity in the Conclusions section (l. 319ff):
"However, the high ice nucleation efficiency of these particles could not be fully explained by the ice nucleation activity of individual organic compounds commonly found in plant tissue, potentially indicating a contribution from primary biological particles or organics associated with microbial activity."

*General Comment 2: […] b) the amount of plant derived INP in the atmosphere is derived without properly motivating all the parameters used in the calculation, which, however, will largely influence the atmospheric importance of these respective INP.*

We mention in our manuscript that our estimates rely on ground-level particle concentrations and therefore we also caution against over-interpreting our results because of "emission fluxes of organic particles acting as INPs being poorly constrained and more detailed modelling case studies needed" (l. 296f).

To add more detail to the description of INP emissions from agricultural areas we have added the following paragraph:
"Anthropogenic dust sources contribute roughly 25 % to the global dust burden, with regional variations ranging from 7 to 75 % (Ginoux et al., 2012). In areas with intense agricultural land use, e.g. in eastern North America, India, eastern China, and Europe, anthropogenic dust emissions contribute generally more than 60 % to the total dust burden (Huang et al., 2015). Note, however, that there is a substantial uncertainty regarding the number and size of particles emitted from agricultural as well as their

transport to cloud altitudes and the resulting atmospheric lifetime. This uncertainty is rooted in a lack of emission flux data above 5-10 m which is the height at which dust fluxes from agricultural areas are commonly observed, e.g. in the study by Zobeck and Van Pelt (2006)." (lines 284ff)

Similarly, for the leaf litter aerosol we have added more information about the studies that we are referencing:
"Sánchez-Ochoa et al (2007) use cellulose found in aerosol particles as a proxy for plant debris concentrations, relying on observations across 6 European sites for a time span of two years, and with two of the sites being located on mountains. Hildemann et al. (1996) used higher alkanes (e.g. ocurring in plant waxes) to fingerprint plant debris in aerosol particles sampled in the greater Los Angeles Area." (lines 277ff)

Additionally, in Fig. 3 we have now scaled down the INP concentrations from agricultural dusts by a factor 100, respectively by a factor 10 for leaf litter to at least partially account for transport losses (see vertical profiles of dust concentrations in Hoose et al., 2010). The surface area values used to convert mass concentrations into aerosol surface concentrations are taken from the BET measurements conducted as part of this study (see Fig. 2) – this has been clarified in the text now, too (lines 292f).

*General Comment 3: More specifically, it has to be said that throughout the text, there is a somewhat strange shortage of mentions of biogenic INP, which, based on their nature, have to be connected to the herein examined plant material samples.*

Please see reply to General Comment 1.

*General Comment 4: In the absence of any physical or chemical test that could have altered the respective samples (e.g., heating, treatment with H2O2), there is no information about the nature of the INPs, therefore any observed activity in the agricultural sample and the leaf litter may originate from the sample material itself or from micro-organisms connected to it (bacteria, fungi, lichen). The overall text has to be revised in connection to this issue.*

Please note that we refer to the potential contribution of primary biological particles in Fig. 2 and the discussion of these results (l. 226ff). We have re-iterated this point in the Conclusions (l. 319ff).

*General Comment 5: Also, for all of those samples from plant-related organic compounds, the sources for the cited quantities for agricultural dust and leaf litter have to be explained in more detail. These numbers are used for a very general calculation afterwards, but the reader needs to know if the given concentrations are valid only locally or for a larger area or worldwide and in which altitudes. A mentioning of "occasionally for very strong wind events" (line 256) rather gives the impression that these values are of no general use in this context, and more information is needed.*

Please see reply to General Comment 2.

*General Comment 6: Also some mentioning of the processes that make particles from dust and leaf litter airborne should be added. These issues and others to be found in my comments below makes a thorough revision of the text necessary before it can be accepted for publication in ACPD.*

We have now added another paragraph to the introduction (l. 54ff):
"Agricultural areas may contribute between 7 and 75 % to the regional dust burden (Ginoux et al., 2012) due to emissions driven by wind erosion and land management activities such as tilling and harvesting

(Funk et al., 2008). Vegetated areas may be another source for complex organic aerosol particles associated with leaf detritus (Coz et al., 2010)."

*Specific comments*

*line 46: "biological aerosol particles" come a bit out of nowhere, here, and it would be good to shortly mention first, that these, too, can be efficient INP (as done for the mineral dust in the previous sentence).*

This sentence now reads:
"Cloud-level concentrations of *potentially very ice-active primary* biological aerosol particles (Hoose and Möhler, 2012) are much lower than background concentrations of mineral dust, with differences of up to 8 orders of magnitude in some cases (Hummel et al., 2018)."

*line 66: Your sentence "decaying plant material being one of the major sources of these macromolecules (Hill et al., 2016)," gives a somewhat wrong impression, as (from what I understood from the literature you cite here) these macromolecules may also originate from biological entities. In their abstract, it says: "Organic INPs . . . may originate from decomposing plant material, microbial biomass, and/or the humin component of the SOM."*

We eliminated "major".

*line 114: It should be added that the identification of an hydrometor as ice is simply based on its size.*

We have added some more information regarding this aspect (please also see l. 152 where this had been mentioned, too).
"Ice crystals are discriminated from droplets by choosing a size threshold which is evaluated individually for each experiment." (l. 124)

*line 167: These individual plant-related organic compounds likely miss the contribution of micro-organisms, so it is not so astounding that they show a comparably low ice activity ("in comparison to samples from natural environments"). A discussion on this is missing completely from the text.*

As mentioned in our reply to General Comment 4, we have acknowledged the potential contribution of primary biological particles, particularly at temperatures above 260 K, in our discussion of Fig. 2. However, now we are also referring to this interpretation at the end of our Conclusions.

*line 175: There is only a VERY low number of points for these "complex polysaccharides" at these higher temperatures (one, one, three, for three different ones of these samples (where the one with the three points seems to consist of two different materials, ALG and PEC)). - Particularly the three dots for ALG/PEC seem to be rather a background, as there is no change in the signal over the temperature range from 254 to 259 K, making this really look like a background issue. Interpreting this as a "weaker dependence on temperature" over-interprets what can be learned from these few data-points.*

We have now pointed out this caveat explicitly. However, we opted for keeping this observation because we would like to highlight the different behavior compared to cellulose which has been investigated in depth. The paragraph in question now read:
"Our data also indicates that the temperature dependence of the polysaccharides investigated in this study is possibly less pronounced than for cellulose. Note that this finding is based only on a few data points due to the low observed ice nucleation efficiency above 252 K." (l. 187ff)

*line 188 ff: The abstract of Conen et al. (2016) says: "Together, both findings suggest that decaying leaves are a strong emission source of IN to the Arctic boundary layer." And they examined litter consisting "of entire leaves and large fragments thereof, mainly from Betula nana and various grasses." – I wonder how this fits to what you wrote: ". . ., leaf litter from the Arctic consisting mainly of grass leaves has been observed to show relatively low ice nucleation efficiencies (Conen et al., 2016), . . ." . Neither seems grass to be the major component of the litter examined by Conen et al. (2016) (or did I overlook something in the Conen-study?), nor do they claim that this is a low ice nucleation efficiency. Temperature wise, data in Conen et al. (2016) only go down to -15◦C, while the temperature range into which your results fall are generally below that, which has to be acknowledged when comparing values. Please revise this part of the text.*

We have now updated this paragraph regarding the sources of leaf litter which also included birch leaves as correctly pointed out by the reviewer. Additionally we have re-phrased this paragraph slightly to further emphasize that we are only comparing ice nucleation efficiencies at 258 K.
"In contrast, at 258 K, leaf litter from the Arctic consisting of birch and grass leaves (Conen et al., 2016) has been observed to show relatively low ice nucleation efficiencies compared to leaf litter in our study based on AIDA results and similar efficiencies when comparing against our droplet freezing assay."

*line 208-209: Is there any explanation why for the two samples discussed above large particles were not a problem and then why they might be a problem here? How else does this one sample differ from these two samples above?*

For the agricultural dust we observed that the suspension contained larger (presumably dust) particles which appeared to sediment relatively fast even though we tried to keep the suspension as well mixed as possible. We did not observe this behavior for the other samples. Hence, we believe that sedimentation might have played a bigger role for the agricultural dust samples.

*212-213: Why should there not also be a contribution from micro-organisms (= biological particles) in the leaf litter sample? After all, originally it was leaves from which Pseudomonas syringae was derived (Maki 1974). While the temperature of the signal's increase does not suggest the influence of a bacteria such as P. syringae, the presence of other micro-organisms could still be responsible for this increase. Testing with e.g., heat or acids or other methods could have increased the understanding.*

We did not intend to exclude a contribution from micro-organisms to the observed ice nucleation efficiency of leaf litter. Based on the reviewer's suggestion we have now included a reference to the study by O'Sullivan et al. (2015) which have investigated the ice nucleation abilities of nanofragments associated with the presence of microbial activity:
"In contrast, the reasons for the steep onset observed for the leaf litter sample are a bit more unclear as most studies investigating primary biological particles have observed freezing onsets and high ice nucleation efficiency already at temperatures above 260 K (see references in Hoose and Moehler, 2012). However, one recent study has found indications for macromolecules associated with microbial activity being ice-active at about 258 K (O'Sullivan et al., 2015)." (l. 228ff)

In our study, we have refrained from conducting heat tests as in the case of complex particles they may produce results which are hard to interpret, i.e. leading a reduction in ice nucleation efficiency in some cases (Suski et al., 2018) but also leading to an increase in the observed ice nucleation efficiency for certain mineral dusts (Boose et al., 2019).

*line 248/249 and line 254 ff: The explanation on how INP concentrations were obtained, i.e., the atmospheric concentrations assumed for leaf litter and agricultural soil dust, needs to be given first, before referring to the derived INP concentrations shown in Fig. 3. Also, it should be discussed in more detail that these indeed are upper boundaries (you mention "very strong wind erosion events"). And also, please motivate the aerosol surface area (1 m2/g). This would need to be the surface area ascribed to leaf litter or to the dust, exclusively. Is this a reasonable value? This needs to be discussed. Also: the mass concentrations: are they to be expected over larger areas, or were these just measured near sources. This is important if you want to make statements on atmospheric wide INP concentrations and the importance of leaf litter and agricultural soil dust in this context.*

As elaborated in our reply to General Comment 2, we have now added more detailed explanations regarding the studies that we used to derive our estimates for ambient plant-related INP concentrations. We hope that it will be now clearer to the reader that these estimates are meant to give order-of-magnitude numbers and that they should be considered as upper limits. Also, we have now added that the assumed surface values come from BET measurements.

*line 265: Could it be that this sentence is based on using unrealistically high atmospheric concentrations of materials that were examined and then using this as an argument for the importance of the present study? For enabling your readers to judge that, motivate the 1 m2/g that you used above, and the mass concentrations of leaf litter and dust, as said above.*

Please see previous reply.

*line 267: The "Section 4: Conclusion" comes a bit abrupt and contains statements that should either be given in a discussion part or otherwise earlier – giving new points of discussion (as the different sources that may emit organic particles or the global extent of the areas that may contribute) is not something that should appear in the conclusions for the first time.*

We agree that this might be confusing for the reader and have eliminated one paragraph (l. 306ff) which contains information some of which has now been moved to the introduction.

*line 277: Why were INSEKT data not also used to obtain atmospheric INP concentrations? As you say here, the measurements with this instrument were done to get results on a wider temperature range, but then they are not used in the further evaluation. Maybe it could also be seen if AIDA or INSEKT give the more trustworthy data? At least this could be discussed when comparing to the data from literature.*

We are unsure to which lines the reviewer is referring here.

Both methods (i.e. AIDA and INSEKT) definitely each have their own value in better understanding ice nucleation, in particular as they allow to investigate different temperature ranges. There are two main aspects which contribute to differences in the observed INAS density values when comparing AIDA experiments against INSEKT droplet freezing studies. First, the method of particle dispersion is different (suspension vs. dry dispersion) and also the surface area used for normalization is different. However, it is not a priori clear how large the differences between the two methods are which is the reason why a direct comparison is still informative to the reader. Secondly, even if we don't look at absolute values when comparing the two methods, INSEKT delivers complimentary information, e.g. by capturing the steep onset at 260 K. In conclusion, both methods are trustworthy, and they deliver individual perspectives on the ice nucleation properties of different samples.

*line 282-284: This sentence might have to be revised if the used values for mass concentration and aerosol surface area are revised or cannot be well justified.*

Please see reply to General Comment 2 – we have now given more detail regarding the underlying assumptions.

*line 284-286: Here it is again not clear why macromolecules from micro-organisms were not considered, as they provide an obvious explanation.*

In this study, we have focused our investigation on organic components in plants as a first step. We hope that our study can be a starting point for investigating organic constituents of other complex particles in the future, e.g. macromolecules associated with the microbial degradation of plant material as suggested by the reviewer.

*Figure 3: As also already mentioned above, I wonder if you want to imply here that the atmospheric INP mostly come from leaf litter and agricultural dust? There is a lot of literature around that ascribes ice activity at temperatures of roughly < -20◦C to desert dust, so I wonder if you really want to challenge this. If you feel your data is strong enough to do that, come up with a good justification. If you don't trust your derived concentrations so much, make it clearer in the text that you present the absolute possible maximum and that likely values from the lower end of the ranges you give or even blow are more likely.*

We would like to emphasize that we have not inferred from our measurements that plant-related aerosols are the most dominant source of INPs. In our study we only wanted to highlight that INPs from vegetation and agricultural areas might be a significant contributor in certain contexts, i.e. certain seasons or regions: "…aerosolized particles from leaf litter and agricultural areas are potentially important contributors to atmospheric INPs." (l. 317)

*Technical comments*

*line 51-52: Check the style of the citations (no brackets should be used in a bracket)*

This inconsistency has been corrected.

*Table 1: Somehow a "555" shows up at the right side of the table. Probably a line number gone wild?*

Unnecessary line numbers have been removed.

*Figure 3: "." is missing at the end of the caption*

Captions have been updated accordingly.

***Response to Reviewer 2***

*General Comment 1: […] First, ice nucleating active (INA) bacteria were identified decades ago and it is well known that vegetation and leaf litter – depending on type – can host dense populations of these bacteria. This component is discussed in the introductory materials, but not brought up again in comparison with the results for vegetation samples. Why not study P. syringae (as a model for this component) with the same systems and compare to that?*

We do agree that a more thorough comparison between various methods to investigate the ice nucleation properties of relevant aerosol species is needed, building on recent intercomparison studies, e.g. for illite and cellulose. For this study, we would like to maintain our focus on plant-related organics, even though we agree that primary biological particles could play an important role for ambient aerosol particles. As discussed later on in this reply, we try to investigate the properties of very "simple" systems as a first step, inviting more complexity in future studies. Also, we would like to highlight that we do compare against the ice nucleation ability of bacteria in Fig. 2.

*General Comment 2: Further, the INA component in bacteria is a lipoglycoprotein (with a particular structure that enables its activity), which presumably inspired some of the choices in Table 1. But this is not explained; and in any case, this is also already well known, so it is not clear what was to be accomplished through the selections made for study unless it is implied that other proteins, lipids, etc. might have IN activity as well (if so, why?). The idea of other "unknown" organic constituents being important (e.g., the macromolecules proposed in earlier work by other groups) is certainly raised, but is not explicitly investigated here – except perhaps by ruling out activity from larger particles composed of the selected compounds.*

Lipoglycoproteins seem like a new promising avenue for future studies. However, in this study we wanted to investigate organic compounds which are major constituents of plant tissue, deliberately not taking into account the impact from microbial degradation processes, in order to be able to investigate very simple systems which we can then contrast with "real" particles. The organic compounds that we investigated in our study are chosen according to the main components of organic matter found in plant tissue (except for cellulose which has been investigated in great depth already).

*General Comment 3: Second, the argument is made that using commercially-available components is preferable because "many of the extraction methods for organic matter may cause significant changes in the physicochemical properties of the extracted organic compounds". Why is this not true also for the commercial products? There is no discussion of how these are manufactured, which seems to be important for the proteins in particular if they are to be considered analogs for natural components.*

It is certainly true that industrial extraction processes will also cause structural changes in the organic compounds. However, we preferred the commercial products because we were not sure how reproducible the extraction process would be and using commercial products allowed us to have larger sample amounts to be available, potentially allowing for follow-up studies and more detailed intercomparisons in the future. The samples investigated in our samples should be considered as analogues but the variability, e.g. across different sources of lignin, remains to be investigated.

We have now separated the sentence in question:
"We used commercially available organic compounds as analogues for plant-derived organics. Note that many of the extraction methods for organic matter may cause significant changes in the physicochemical properties of the extracted organic compounds (Kögel-Knabner, 2002)."

*General Comment 4: I also have questions regarding the process for generating particles of carnauba wax, which was the only component identified as having significant IN activity: on line 216 it is stated that, "Unfortunately, it was not possible to reliably determine INAS density values for carnauba wax (LIP) due to its very low dispersibility." It is appreciated that generating reproducible particles from solid samples is very difficult, but the uncertainties associated with this should be quantified and carried through the analyses. The results for carnauba wax are noted to be surprising (lines 284-285) but few fully satisfying reasons for this result can be deduced from the present study (some ideas are presented in lines 177-185).*

The issue with the carnauba wax was that is consisted mainly of larger particles which was less problematic for dry dispersion (rotating brush generator and cyclone impactors) than for the creation of suspensions where we would have needed to grind the particles, resulting in a substantially different particle distribution and potential surface effects from the grinding procedure.

*General Comment 5: A third major point with regard to atmospheric implications is that while soils, leaf litter, harvest debris, etc. can have high densities of INA bacteria or other ice-active components, the mobilization of particles containing those components into the boundary layer, and further, to altitudes where they can impact cloud formation, is a different matter. Limited prior studies suggest there is no direct relationship between surface concentrations and atmospheric concentrations and the atmospheric concentrations become relevant only under conditions where the surface is strongly disturbed (as alluded to in the text). Thus the implications of any findings with respect to atmospheric processes have to be tempered by this consideration. In particular, the concluding sentence of the Abstract, "In contrast, complex biological particles may exhibit ice nucleation activities which are up to two orders of magnitude higher than observed for cellulose, making ambient plant-derived particles a potentially important contributor to the population of ice-nucleating particles in the troposphere" is not a unique conclusion from this work but has been suggested previously, and needs to be modified to acknowledge that the relationship between the surface and ambient concentrations needs to be better understood before quantifying the importance of this source on regional and global scales.*

The last sentence of the abstract now reads:
"In contrast, complex biological particles may exhibit ice nucleation activities which are up to two orders of magnitude higher than observed for cellulose, making ambient plant-derived particles a potentially important contributor to the population of ice-nucleating particles in the troposphere, *even though major uncertainties regarding their transport to cloud altitude remain*."

We have also adjusted the atmospheric INP concentrations represented in Fig. 2 by applying a scaling factor of 10 for leaf litter emissions and a factor of 100 for the agricultural dust emissions, in order to account for transport losses (see vertical profiles for dust in Hoose et al., 2010). Due to the probably episodic character of these emissions, our estimates still need to be considered as upper limits.

*Line 206: I have additional comments for consideration, as follows. It is stated that for some of the tested samples, the AIDA and microdroplet methods agree (lines 206-208). However, there is no overlap*

*between these methods, and the surface area determinations use very different approaches, calling this agreement into question.*

For the leaf litter sample, there is an overlap at around 257 K. For the plant protein, there is no overlap but the trajectories are reasonably close, so that we can assume that these trajectories can be virtually extrapolated by 1 K. Also, Fig. 2 intends to highlight that agreements and differences between two ways of inferring INAS densities are strongly dependent on aerosol types.

*Line 107: The particle background concentrations for AIDA are stated (line 107) as 100 L-1. Comparing to Figure 3, I'm unclear how this is taken into account; the x-axis scales on Figures 1 and 2 are different, indicating that AIDA is limited at the warmer temperatures, presumably due to this background?*

Yes, droplet freezing measurements can be conducted in a way that they are more sensitive at warmer temperatures (i.e. by choosing the weight percentage of suspended particles) and are therefore a great complement to AIDA experiments. For samples like agricultural dust, it can be difficult to reach particle concentrations which allow for ice crystal concentrations above background to be observed during AIDA experiments at very small supercoolings. One of the reasons can be the availability of fine material as we use cyclone impactors to eliminate larger particles and typically the sample amount is limited. For Fig. 3, we only applied the parameterization that we derived from AIDA experiments to prescribed ambient aerosol concentrations. The background limits therefore don't need to be considered.

*Line 237: Prior work by Hiranuma et al. (2015b) is cited for data on cellulose for comparison to the present work. The intercomparisons published by Hiranuma et al (2019) are also cited, however, in that study, it is noted that "While the diverse instruments employed in this study agree in that cellulose has the capacity to nucleate ice, their quantitative agreement is poor. Unfortunately, it is not possible yet to say what the cause of this disagreement is." Does this statement apply to the two techniques used in the manuscript? Hiranuma et al. (2019) also call for "comprehensive studies on the ice nucleation activity of other important plant structural materials, such as cellulose polymorphs, lignin materials, lipids, carbohydrates and other macromolecule saccharides", so the present study is a nice follow-on to that recommendation. However, the issue of whether follow-on studies are premature at this point, if there are fundamental questions regarding the measurements and their interpretation, needs to be addressed.*

There are differences between the two methods which are challenging to resolve, e.g. the role of soluble material which might lead to differences between wet and dry dispersion. Nevertheless, we want to consent to the point that the reviewer made based on the findings in Hiranuma et al. (2019), calling for more comprehensive studies to better understand differences between methods are dependent on the aerosol type being investigated. We consider our study as a first step highlighting differences between organic components and calling for more detailed studies of this subject.

*Line 89: "ambient samples from vegetated environments": my comments above assume these are bulk samples and not obtained by filtering of ambient air. If my interpretation is correct, perhaps the language here needs to be clarified.*

The word "ambient" was chosen to clearly distinguish these samples from the individual components and thus refers to samples from the outside environment. We have tried to be a bit more mindful with our choice of words and have therefore adjusted the text by either adding the word "bulk", eliminating

the word "ambient" or by replacing "ambient" with "from vegetated and agricultural sources" where appropriate.

*Line 148: Brunauer is misspelled. The uncertainties introduced by the different estimates of surface area should be more thoroughly discussed and represented in the figures (how are the uncertainty bars in the figures computed – is this from the variation in the repeat experiments, or does it include other considerations such as surface area?)*

Spelling has been corrected.

For the AIDA experiments, there are two factors contributing to the error bars as displayed in Figs. 1 and 2: the uncertainty of the observed ice crystal concentration $\Delta n_{ice}/n_{ice} = 0.2$ and the uncertainty of the aerosol surface area concentration $\Delta A_{aer}/A_{aer} = 0.35$ (see line 152ff). We have now added one more sentence in the Methods section to make it clearer that the aerosol surface area used to derive the AIDA based INAS density values relies on a geometric surface area estimate:

"The combined aerosol size distributions are used to estimate the available aerosol surface based on volume-equivalent sphere diameters which then results in an estimate of the geometric surface area." (line 117ff)

For the INSEKT measurements, the error bars are determined by the statistical uncertainty of the number of frozen droplets (which translates to INP concentrations) and the uncertainty in measuring the BET surface area. However, in most cases the uncertainty of the BET measurements is less than 10 % which is smaller than the statistical uncertainty. We have now slightly updated this paragraph (l. 157ff): "For our INAS density uncertainty analysis, we considered only the uncertainty of the cumulative INP concentrations *which is based on statistics*. Confidence intervals (at 95 %) have been estimated according to the improved Wald interval which implicitly assumes a normal approximation for binomially distributed measurement errors (Agresti and Coull, 1998). Hence, in our INAS density analysis, we neglected the uncertainties of the BET surface measurements which are in most cases considerably smaller *(i.e. $\Delta A_{aer}/A_{aer} < 0.1$)* than the previously described *statistical* uncertainties of the cumulative INP concentrations (Hiranuma et al., 2015a)."

We have also added one more aspect regarding the sources of differences between INSEKT and AIDA derived INAS densities (l. 167):
"Additionally, suspending particles in water may lead to the desorption and potential redistribution of soluble material. This change in soluble material could also lead differences in the observed ice nucleation properties when comparing cloud chamber experiments with droplet freezing studies."

*Line 174: Desert dust (Ullrich et al., 2017) is mentioned for comparison, but not shown?*

We did not want to overload Fig. 2 – in order to make it more obvious to the reader, we have now added a note saying the data from Ullrich et al. (2017) is not displayed in Fig. 2.

*Line 204: Is the background for the microdroplet method shown here or in another publication?*

Another publication explaining these details is in preparation – pure water droplets commonly start freezing at temperatures below 249 K.

*Line 110, 148: could these aerosol size and surface area distributions be shown in the Supplementary Material? This is potentially useful information for other studies that might seek to explore similar science questions with other techniques.*

We have now added size distributions for lignin and leaf litter to the Supplementary Material.

**References**

Boose, Y., P. Baloh, M. Plötze, J. Ofner, H. Grothe, B. Sierau, U. Lohmann, and Z. A. Kanji. 2019. "Heterogeneous Ice Nucleation on Dust Particles Sourced from Nine Deserts Worldwide-Part 2: Deposition Nucleation and Condensation Freezing." *Atmospheric Chemistry and Physics* 19 (2): 1059–76.

Coz, E., B. Artíñano, L. M. Clark, M. Hernandez, A. L. Robinson, G. S. Casuccio, T. L. Lersch, and S. N. Pandis. 2010. "Characterization of Fine Primary Biogenic Organic Aerosol in an Urban Area in the Northeastern United States." *Atmospheric Environment* 44 (32): *3952–62.*

Funk, R., H. I. Reuter, C. Hoffmann, W. Engel, and D. Öttl. 2008. "Effect of Moisture on Fine Dust Emission from Tillage Operations on Agricultural Soils." *Earth Surface Processes and Landforms* 33 (12): 1851–63.

Ginoux, P., J. M. Prospero, T. E. Gill, N. C. Hsu, and M. Zhao. 2012. "Global-Scale Attribution of Anthropogenic and Natural Dust Sources and Their Emission Rates Based on MODIS Deep Blue Aerosol Products." *Reviews of Geophysics* 50: RG3005.

Hoose, C., J. E. Kristjánsson, J.-P. Chen, and A. Hazra. 2010. "A Classical-Theory-Based Parameterization of Heterogeneous Ice Nucleation by Mineral Dust, Soot, and Biological Particles in a Global Climate Model." *Journal of the Atmospheric Sciences* 67 (8): 2483–2503.

Hoose, C., and O. Möhler. 2012. "Heterogeneous Ice Nucleation on Atmospheric Aerosols: A Review of Results from Laboratory Experiments." *Atmospheric Chemistry and Physics* 12 (20): 9817–54.

Huang, J. P., J. J. Liu, B. Chen, and S. L. Nasiri. 2015. "Detection of Anthropogenic Dust Using CALIPSO Lidar Measurements." *Atmospheric Chemistry and Physics* 15 (20): 11653–65.

Möhler, O., D. G. Georgakopoulos, C. E. Morris, S. Benz, V. Ebert, S. Hunsmann, H. Saathoff, M. Schnaiter, and R. Wagner. 2008. "Heterogeneous Ice Nucleation Activity of Bacteria: New Laboratory Experiments at Simulated Cloud Conditions." *Biogeosciences* 5 (5): 1425–35.

O'Sullivan, D., B. J. Murray, J. F. Ross, T. F. Whale, H. C. Price, J. D. Atkinson, N. S. Umo, and M. E. Webb. 2015. "The Relevance of Nanoscale Biological Fragments for Ice Nucleation in Clouds." *Scientific Reports* 5 (January): 8082.

Polen, M., E. Lawlis, and R. C. Sullivan. 2016. "The Unstable Ice Nucleation Properties of Snomax® Bacterial Particles." *Journal of Geophysical Research: Atmospheres* 121 (19): 11,666–11,678.

Zobeck, T. M., and R. S. Van Pelt. 2006. "Wind-Induced Dust Generation and Transport Mechanics on a Bare Agricultural Field." *Journal of Hazardous Materials* 132 (1): 26–38.

---

## Author Response (AR2)

**Final response: Complex plant-derived organic aerosol as ice-nucleating particles – more than a sum of their parts?**

Dear referee, we thank you for reviewing our revised paper and for accepting it with minor revisions. In the following we reply to your points:

*1) In the abstract, while cellulose is cited as "has been suggested by recent studies as a proxy for quantifying the primary cloud ice formation caused by particles originating from vegetation", it is still not mentioned at all that biological particles are likely to play a much larger role, at least at higher freezing temperatures (> 258 K, maybe even further down). Your examinations go up into this temperature range and therefore mentioning them in the abstract helps to prevent that readers of your text in the future will ignore these important contributors so atmospheric INP.*
We have added to the second paragraph in the abstract which now reads:
„In this work, we present results from laboratory studies investigating the immersion freezing properties of individual organic compounds commonly found in plant tissue and complex organic aerosol particles from vegetated environments, without specifically investigating the contribution from biological particles which may contribute to the overall ice nucleation efficiency observed at high temperatures."
In addition, we have substituted "biological" with "biogenic" in line 35.

*2) The new text starting in line 58 ("Agricultural …") should be started a bit differently so that it fits to the flow of the text before. The motivation here is that this "missing source of INPs …" could be found in agricultural dust, as far as I understand it.*
The beginning of this sentence now reads:
„One of the potential sources for these terrestrial INPs are agricultural areas…''

*3) Line 121: Add "area" between "surface" and "based".*
Done.

*4) The new text starting in line 145 fits better to the next section (2.3), as this is important for the determination of the surface area.*
We have moved the sentence in question.

*5) Line 169 ff: You argue that sedimentation is negligible. But later in the text, for one sample, you even argue with sedimentation to explain the observed difference. And I agree with your later argument. But that also means that issues with a wrongly estimated surface area can be expected to contribute more to potential deviations between different methods than the redistribution of soluble material. In order to not give a wrong impression, "negligible" should not be said here! I suggest to delete the part of the sentence with "negligible" in it and reformulate it to -> "But to understand the full effect, more future studies are needed."*
We have re-phrased as suggested.

*6) Line 161: Add a "to" between "lead" and "differences".*
Done.

*7) Line 297-298 and Fig. 3: Do you mean Fig. 3 in line 297? (Otherwise I am confused and revisions in the description of Fig. 2 and related text would be needed.)*
This has been corrected.
*Even if you mean Fig. 3 here, I am still a bit concerned as this scaling down seems a bit arbitrary. And Fig. 3 has not changed between the first submission and now, so I wonder if this scaling down was already done in the first version, or if Fig. 3 has really not changed? Are these scaling numbers based*

*on any scientific evidence that can be cited here? That would improve the message of the work substantially.*

The difference is very hard to notice but in Fig. 3 the blue area has moved down between the two versions. The one change which we highlighted in the track changes document but did not mention in the reply was that we now use the measured BET surface areas instead of a uniform value of 1 $m^2/g$ which we initially chose because the surface area of ambient aerosol from terrestrial sources is highly uncertain. We have now added our rationale for the scaling factors:

[revised manuscript text omitted]